# Out-of-equilibrium microcompartments for the bottom-up integration of metabolic functions

Thomas Beneyton[1], Dorothee Krafft[2], Claudia Bednarz[2], Christin Kleineberg[2], Christian Woelfer[2], Ivan Ivanov[2], Tanja Vidaković-Koch[2], Kai Sundmacher [2,3] & Jean-Christophe Baret [1]

Self-sustained metabolic pathways in microcompartments are the corner-stone for living systems. From a technological viewpoint, such pathways are a mandatory prerequisite for the reliable design of artificial cells functioning out-of-equilibrium. Here we develop a microfluidic platform for the miniaturization and analysis of metabolic pathways in man-made micro-compartments formed of water-in-oil droplets. In a modular approach, we integrate in the microcompartments a nicotinamide adenine dinucleotide (NAD)-dependent enzymatic reaction and a NAD-regeneration module as a minimal metabolism. We show that the microcompartments sustain a metabolically active state until the substrate is fully consumed. Reversibly, the external addition of the substrate reboots the metabolic activity of the microcompartments back to an active state. We therefore control the metabolic state of thousands of independent monodisperse microcompartments, a step of relevance for the construction of large populations of metabolically active artificial cells.

---

[1] CNRS, Univ. Bordeaux, CRPP, UMR 5031, 115 Avenue Schweitzer, 33600 Pessac, France. [2] Max Planck Institute for Dynamics of Complex Technical Systems, Sandtorstrasse 1, 39106 Magdeburg, Germany. [3] Otto-von-Guericke University, Process Systems Engineering, Universitätsplatz 2, 39106 Magdeburg, Germany. Correspondence and requests for materials should be addressed to J.-C.B. (email: jean-christophe.baret@u-bordeaux.fr)

Metabolic activity is a hallmark of living systems. The chemical transformations of molecules present in the environment provide the energy required for the cell to maintain its out-of-equilibrium state and thereby prevent its decay toward a state of minimal energy[1]. The complexity of metabolism in living systems is a widely accepted feature[2]. Attempts to modify or simplify these complex networks of reactions—the task of metabolic engineering—face the problem that any modification within these networks affects the overall behavior of the cell[3]. While metabolic engineering is centered on the genetic modification of living cells[4], a bottom-up approach to build controlled cell-like systems from soft matter constituents by integrating synthetic pathways in microcompartments has become a promising alternative[5,6]. The organization of metabolic processes in microcompartments is a mandatory prerequisite for the construction of artificial cells[7,8]. Realizing such an assembly in a bottom-up approach would provide an unprecedented level of control on the constituents of the metabolic functions and allow a fine control of its waste and side products. To date, the bottom-up approach for the creation of life-like artificial microsystems is still in its infancy[9–14] but key building blocks are step by step assembled[15,16], from the creation of microcompartments[17,18] to the in vitro integration of complex artificial metabolic pathways[19]. Under the assumption that primary living cells have emerged from prebiotic systems made of soft matter[6], one should be able to integrate elementary metabolic activities in minimal systems to maintain these systems out-of-equilibrium. Under these conditions, even simple reactions are relevant in a protocell context, as evidenced by recent models of protocells divisions based on metabolic activity[20].

Microfluidics has become a key technology for the creation, manipulation and analysis of microcompartments, as well as for the control and programming of in vitro biochemical processes[21–23]. Microfluidics provides means to quantitatively manipulate minute volumes of biological materials in the form of soft-matter systems such as (multiple) emulsions or vesicles[24–26]. The technology enables the miniaturization and parallelization of assays for high-throughput biological experiments in protein engineering[27], cell screening[28,29], molecular diagnostics[30], or sequencing[31–33]. The same tools used to perform complex chemical assays and analysis are now integrated to build-up and analyse large population of artificial microcompartments having functions and properties mimicking those of living systems[18,34,35].

Here we use microfluidic systems for the integration of minimal metabolic reactions in man-made microcompartments. We design a platform for the production, manipulation and analysis of millions of individual monodisperse microcompartments in a water-in-oil emulsion. We develop an assay based on nicotinamide adenine dinucleotide (NADH) fluorescence to quantify the metabolic state of the microcompartments. The minimal metabolism is constructed from a reaction converting glucose-6-phosphate (G6P) into 6-phosphogluconolactone (GLP). The reaction is catalysed by glucose-6-phosphate dehydrogenase (G6PDH), an enzyme involved in the pentose phosphate pathway[36]. A key feature to integrate is the ability to function under conditions where the reaction is sustained independently of the cofactor stoichiometry. Here, the full conversion of the metabolic substrate requires the regeneration of the cofactor $NAD^+$. The regeneration module is made of inverted membrane vesicles (IMVs) extracted from E. coli. We monitor the state of the microcompartments (active vs sleeping) by the readout of the microcompartments fluorescence. We analyze the kinetics of the system both in bulk experiments and over thousands of monodisperse microcompartments. A sustained active state is maintained for times varying between several minutes to hours depending on the experimental conditions, such as the concentrations of the IMVs and of the substrate. The decay of the active state is determined by the initial amount of the substrate in the microcompartments. We show that the active state is recovered by the injection of fresh substrate in the microcompartments, using a high-throughput targetted delivery of substrate in each droplet[37]. This reboot of activity actually confirms that the end-state upon full substrate consumption is a sleeping state that can indeed be reactivated. Our experiments therefore provide quantitative measurements of the metabolic state of the microcompartments, a measurement of these states over thousands of microcompartments and the on-and-off switching of the metabolic activity of the microcomparment through a chemical regeneration, all elementary steps required for the construction of autonomous metabolically active artificial cells.

## Results

**Compartmentalized metabolic reactions in microfluidics**. We use water-in-oil (w/o) droplets stabilized by a block-copolymer surfactant as artificial microcompartments to host enzymatic reactions involved in metabolic pathways (such as the pentose phosphate pathway or the Krebs cycle). W/o droplets offer a powerful means of compartmentalization and they are produced, manipulated and analyzed at very high-throughput using droplet-based microfluidics[24]. Droplet-based microfluidics provides both control of individual droplets and the capacity to process millions of them in automated and parallel processes. Although, the w/o interface acts as an almost impermeable membrane, a mediated transport of chemicals can be induced, either passively through the external oil phase[38] or actively using microfluidic injection techniques[37,39] to precisely control the composition of the microcompartments.

The encapsulation and monitoring of the metabolic cascades are performed using three droplet-based microfluidic platforms implemented depending on the experimental needs (timescale of the assay, multiplexing) based on previously reported systems (Supplementary Fig. 1–3 and Methods section). The biochemical measurements are based on a high-throughput fluorescence readout at the single-droplet level. One of the important features of the platforms is the preparation of emulsions in a multiplexed format which provides means to analyse several experimental conditions—including the relevant controls—within the same experiment. The different compositions are encoded using a fluorescent dye as a marker to automatically trace back the experimental conditions in the droplet at the time of the readout of the assay according to previously published procedures[30,40–42]. We generate either 1-bit, 4-bit or 8-bit emulsions when one, four or eight experimental conditions are simultaneously assayed on chip. The optical setup for the fluorescence analysis of the microcompartments is described in Supplementary Figure 4, while the fluorescence spectrum of all fluorophores used in this work are shown in Supplementary Figure 5. The typical workflow is described in Fig. 1a. In brief, the metabolic modules (i.e., enzymes and proteins) are encapsulated in 30 pL w/o droplets. The microcompartments are activated by the picoinjection of a metabolic substrate or cofactor[37]. The droplets are incubated on-chip in delay-lines[43]. Reaction kinetics is monitored over time in the microcompartments using fluorescence measurements at specific incubation points. The system is calibrated for all the fluorophores used here using 8-bit emulsions (Supplementary Figure 6 and Supplementary Note 1). The fluorescence intensity is found to be proportional to the fluorophore concentration in the range of used concentration (typically up to 1 mM).

Using this workflow, the kinetics of the compartmentalized enzymatic reactions are monitored within large populations of microcompartments to generate statistically relevant data with a

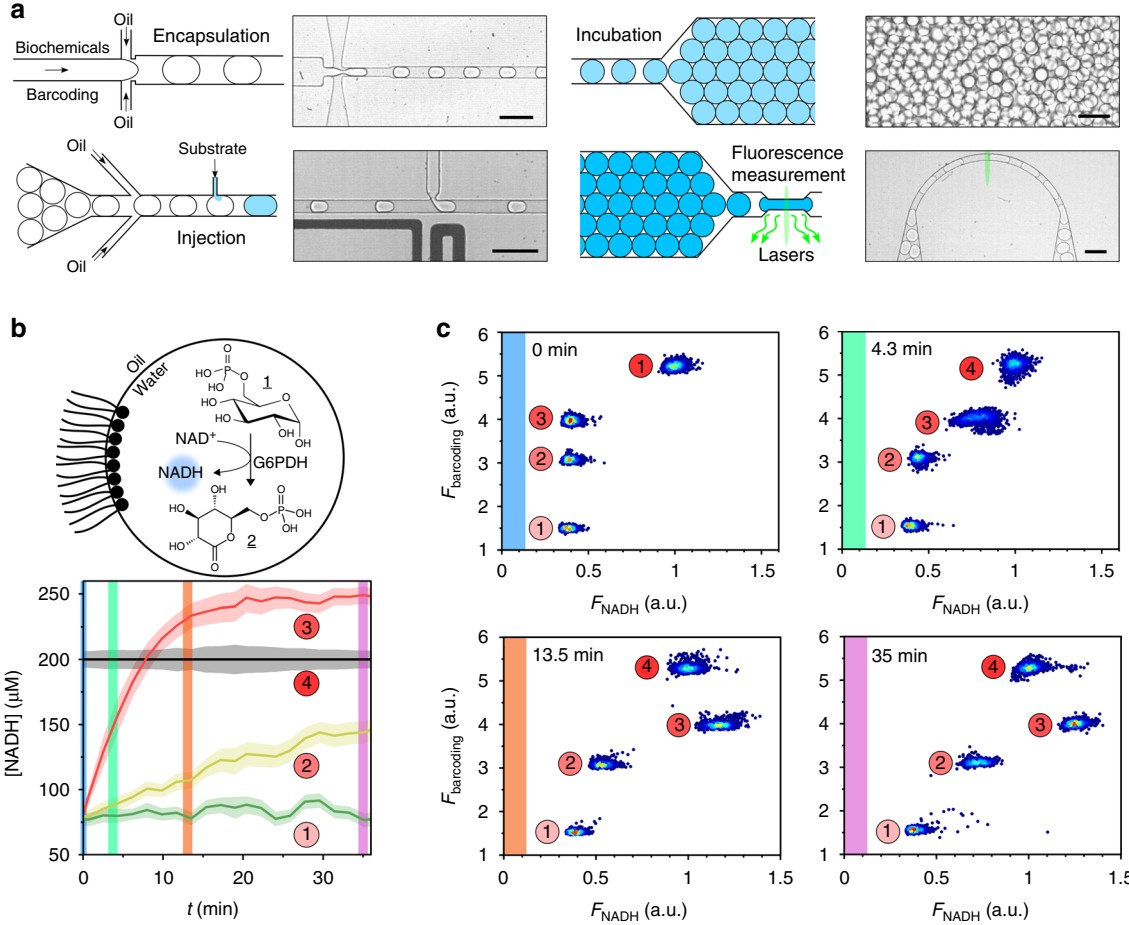

**Fig. 1** Microfluidic platform for monitoring compartmentalized metabolic reactions. **a** Microfluidic workflow. Biochemical components are encapsulated in 30 pL w/o droplets stabilized by a block-copolymer surfactant. Different droplet compositions are barcoded with a fluorescent dye. Compartmentalized reactions are activated by picoinjecting a metabolic substrate or cofactor. The microcompartments are incubated on-chip to monitor their metabolic state using fluorescent readouts. Scale bars 100 μm. **b**, **c** Kinetics of compartmentalized reactions. **b** G6PDH activity. Glucose-6-phosphate dehydrogenase (G6PDH) oxidizes D-glucose-6-phosphate 1 (G6P) into 6-phospho-D-glucono-1,5-lactone 2 (GLP) with the concomitant reduction of NAD+ into NADH. NADH concentration versus time ($t$) of 30 pL w/o droplets containing NAD+ (250 μM) and G6PDH 0, 0.01 and 0.08 U mL$^{-1}$ (green, yellow, and red curves, respectively) or NADH 200 μM internal reference (black) after injection of G6P substrate (1 mM). Error bars are defined as s.d. ($N = 2000$). **c** 2D histograms of NADH fluorescence versus barcoding fluorescence (30, 60, 90, and 120 μM sulforhodamine B, respectively) at different incubation times for the 4-bit emulsion. Reactions are performed in NaOH-Tricine buffer (100 mM, pH 8.0) with MgCl$_2$ 5 mM

resolution down to the single-microcompartment level. The platform is first characterized with a classical fluorogenic assay: the β-galactosidase activity, catalyzing the sequential hydrolysis of fluorescein di(β-D-galactopyranoside) into galactose and fluorescein (Supplementary Figure 7) is used as a control for our system. The kinetics of the compartmentalized reaction is in good agreement with the kinetics measured using standard 384-well plate experiments (Supplementary Figure 7), showing that the workflow proposed here ensures quantitative measurements at the single-droplet level.

In the general case however, implementing a fluorogenic assay is not trivial. It is limited to a small range of highly specific reactions and usually implies a chemical modification of the natural substrate of the enzyme or the use of an additional enzymatic chain reaction that may interfere with the assayed enzyme and affect its native activity. In an attempt to build-up a generic platform, we focus on quantitative measurements of the concentration of a cofactor in the reaction. NAD is a cofactor involved in redox metabolism in all living cells. We use the fluorescence properties of its reduced form (NADH) to monitor NAD-dependent metabolic reactions without designing specific

fluorogenic assays. To demonstrate that our approach is valid, we use three independent systems involving NADH. We consider a single-step reaction and monitor the activity of compartmentalized glucose-6-phosphate dehydrogenase, which oxidizes G6P (1) into GLP (2) with the concomitant reduction of NAD+ into NADH (Fig. 1b, c). The kinetics of the compartmentalized reaction is monitored over time for two enzyme concentrations and shows good correlation with 384-well plate experiments (Supplementary Figure 8). We then demonstrate the general applicability of our systems with two-steps reactions where NADH is involved either in the first or the last step of the cascade, such as the L-malate dehydrogenase/citrate synthase sequence, which is part of Krebs cycle, and the glycerol kinase/ glycerol-3-phosphate dehydrogenase sequence, which is part of glycerol metabolism (Supplementary Figures 9 and 10).

In summary, the developed platform is designed as a generic system and allows for the monitoring of biochemical processes involved in metabolic pathways, with a versatile optical readout of the metabolic state of microcompartments using NADH fluorescence. This high-throughput methodology is then used to monitor the metabolic state of millions of biomimetic

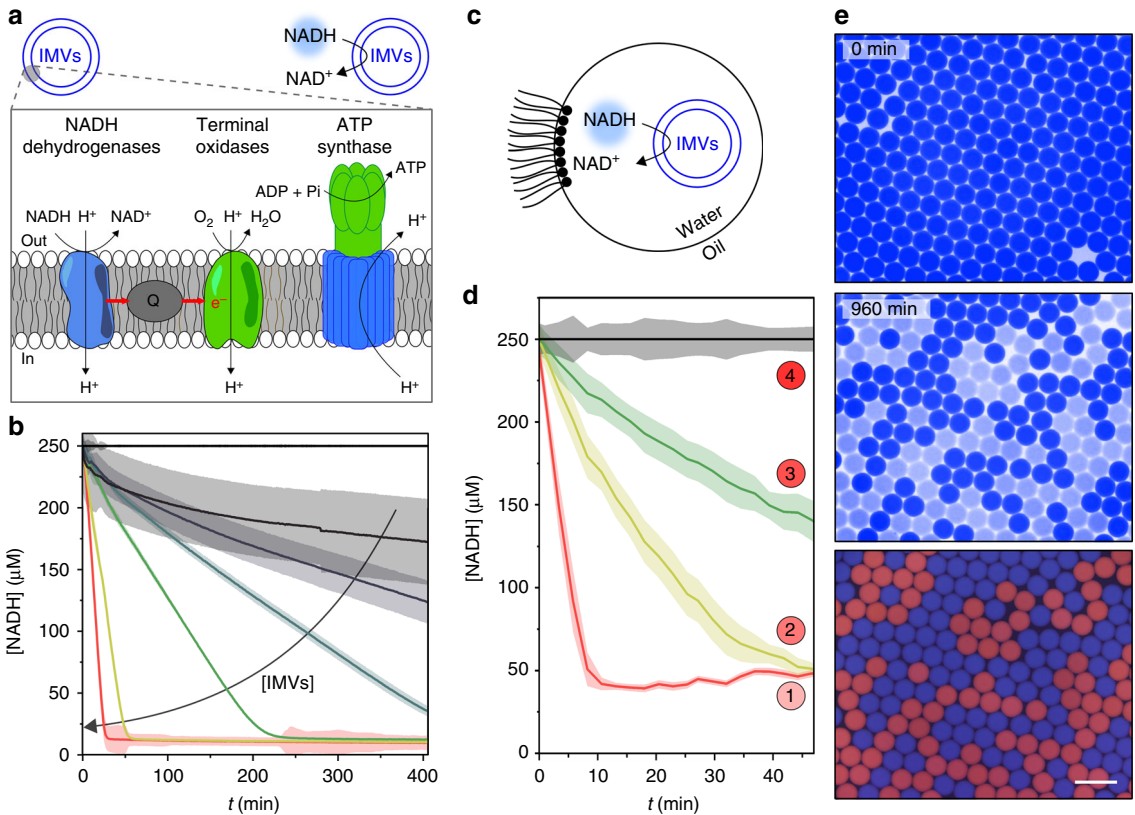

**Fig. 2** NAD cofactor regeneration module. **a** Scheme of inverted membrane vesicles (IMVs). NADH is oxidized by the NADH dehydrogenase activity of the IMVs. The enlargement shows the graphic view of the respiratory chain complexes with ubiquinone Q being reduced by NADH dehydrogenases and oxidized by terminal oxidases complex to finally shuttle electrons to molecular oxygen. The proton gradient generated by this electron transport chain is used by ATP synthase to form ATP via oxidative phosphorylation. **b** NADH concentration versus time ($t$) for increasing IMVs concentrations. Error bars are defined as s.d. ($N = 3$). **c**–**e** IMVs activity within w/o droplets. **c** Graphic view of the system. **d** NADH concentration versus time ($t$) of 30 pL w/o droplets containing IMVs 0, 50, 100, and 300 vesicles per droplet (black, green, yellow, and red curves, respectively) after injection of NADH (250 μM). Error bars are defined as s.d. ($N = 2000$). **e** Blue and red fluorescence micrographs of 300 pL droplets containing NADH (1 mM) only or a mixture of NADH (1 mM), IMVs (100 vesicles per droplet) and sulforhodamine B (20 μM) after 0 and 960 min incubation. Scale bar 200 μm. Reactions are performed in NaOH-Tricine buffer (100 mM, pH 8.0) with MgCl₂ 5 mM

microcompartments based on NAD or NADH-dependent enzymatic activities.

**Module for the regeneration of NAD⁺ cofactor.** Bioactive microcompartments need a constant energy supply in order to stay thermodynamically out-of-equilibrium and be able to activate their metabolism in presence of a given substrate. In the case of NAD-dependent reactions, the constant supply of NADH/NAD⁺ would provide the chemical energy needed for a continuous metabolization of uptaken substrate. In this sense, a self-sustained system is required for the in situ regeneration of the cofactor.

The use of NADH oxidase has been reported as an efficient method to maintain the NADP⁺/NADPH redox balance in synthetic biochemistry systems[44]. Here, we follow a bottom-up strategy for the integration of functional modules based on microcompartments. We use Inverted Membrane Vesicles (IMVs) extracted from *E. coli* as functional microcompartments for the regeneration of NAD⁺ cofactor.

The IMVs contain both the essential electron transport chain proteins of the respiratory chain (NADH dehydrogenase activity)[45,46] and the ATP synthase for oxidative phosphorylation[47]. Apart from active transport studies[48,49], the use of the respiratory chain functionality of IMVs has only been employed for cell-free protein production[47]. In our experiments, we aim at

using the NADH dehydrogenase activity of the bacterial respiratory chain complexes of IMVs as cofactor regeneration module for our enzymatic reactions (Fig. 2a). IMVs are usually obtained by mechanical breakage[50]. The IMVs are extracted from *E. coli* by disintegration of the bacterial membrane and subcellular fractionation (Supplementary Note 2). The resulting small membrane fragments form vesicles with an average size of 167 ± 39 nm (with a volume of ~2 aL) and a typical density of $2.2 \times 10^{11}$ vesicles per mL (Supplementary Figure 11). These vesicles show both ATP synthase (Supplementary Figure 12, Supplementary Note 3) and NADH oxidation activities (Supplementary Figure 13). Here we solely use the NADH oxidation activity for our purposes and the vesicles fully convert NADH into NAD⁺ (Fig. 2b). The rate of the reaction depends linearly on the IMVs concentration (Supplementary Figure 13).

The activity of the compartmentalized IMVs is then measured in microfluidics (Fig. 2c). The IMVs are encapsulated in 30 pL w/o droplets at an average number $N$ of vesicles per compartment ($N \approx 55$, 110 or 330 depending on the dilution factors used). Under the assumption of a uniform concentration of vesicles in the bulk, we expect the number of vesicles in the droplet to be distributed as a Gaussian with a width of order $\sqrt{N}$. The concentration of vesicles in the droplets will therefore fluctuate by at most ±15%. After picoinjection of NADH, the fluorescence of the microcompartments decreases as a function of time as the result of the NADH conversion. The kinetics are comparable to 384-well plate

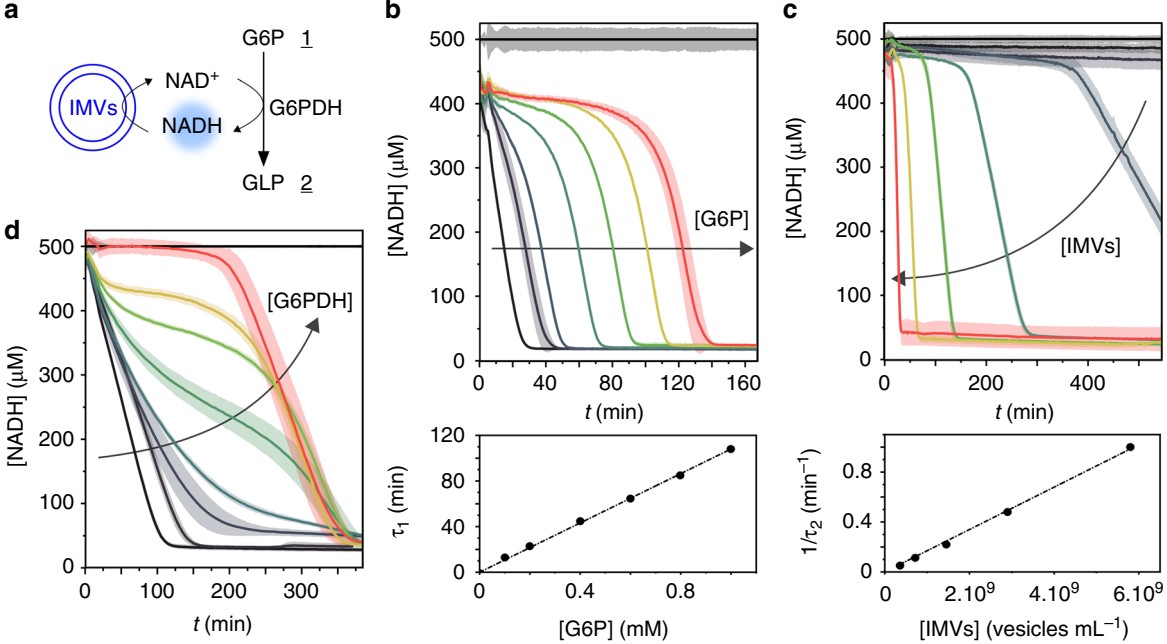

**Fig. 3** Coupling metabolic and cofactor regeneration modules. **a** Reaction network. The G6PDH activity is coupled to the IMVs activity for NADH constant recycling. **b** G6P dependency. NADH concentration versus time ($t$) of solutions containing NADH (500 μM), IMVs ($1.3 \times 10^9$ vesicles per mL), G6PDH (0.1 U mL$^{-1}$) and increasing concentrations of G6P (0, 0.1, 0.2, 0.4, 0.6, 0.8, and 1 mM). Control solution without IMVs in black. The graph below shows the dependency of the lifetime of the reaction ($\tau_1$) toward G6P concentration. **c** IMVs dependency. NADH concentration versus time ($t$) of solutions containing NADH (500 μM), G6P (0.8 mM), G6PDH (0.5 U mL$^{-1}$) and increasing concentrations of IMVs ($0.9 \times 10^7$, $1.8 \times 10^8$, $3.6 \times 10^8$, $7.2 \times 10^8$, $14.5 \times 10^8$, $2.9 \times 10^9$, and $5.8 \times 10^9$ vesicles per mL). The graph below shows the dependency of the lifetime of the reaction ($\tau_2$) toward IMVs concentration. **d** Enzyme dependency. NADH concentration versus time ($t$) of solutions containing NADH (500 μM), IMVs ($5.4 \times 10^8$ vesicles per mL), G6P (0.8 mM) and increasing concentrations of G6PDH (0, 0.0008, 0.004, 0.006, 0.0086, 0.013, 0.02, and 0.5 U mL$^{-1}$). Control solution without IMVs in black. Reactions are performed in NaOH-Tricine buffer (100 mM, pH 8.0) with MgCl$_2$ 5 mM. Error bars are defined as s.d. ($N = 3$)

experiments (Fig. 2d and Supplementary Figure 14) and the increase of the droplet to droplet variability compared to the bulk experiments is likely a consequence of the statistical fluctuations in the number of encapsulated IMVs per droplet. Similarly, the plating of droplets immobilized in a 2D observation chamber (Supplementary Figure 15[29], Supplementary Note 4) and a time-lapse imaging of the emulsion using fluorescence microscopy shows the conversion of NADH in the microcompartments containing IMVs (Fig. 2e, Supplementary Note 5, Supplementary Figure 24).

The complete regeneration of NADH at a concentration of 250 μM in a 30 pL volume requires the transfer of ~$10^{16}$ protons towards the core of the IMVs. For 100 IMVs in a droplet, each IMV would contain ~$10^{14}$ protons in a volume of 2 aL, corresponding to a volume much larger than the volume of an IMV filled with water. This unphysical situation implies that a leakage of protons has to spontaneously occur from the vesicle to the droplet bulk. In living cells, the inward and outward proton fluxes are balanced by the coupling of the respiratory chain with the oxidative phosphorylation of ADP. However, the respiratory chain is also known to be functional even when uncoupled from the ATP synthase activity in a state referred to as basal or state 4 respiration. This state is characterized by the leakage of protons through proton pumps and/or through the membrane[51,52]. From the estimation of the proton flux, our experiments suggest that a similar process occurs with the IMVs.

These results indicate that IMVs are efficient NAD$^+$ regeneration systems in droplet microcompartments. Our next step consists in the coupling of this cofactor regeneration module to a NAD-dependent enzymatic module in order to obtain a self-sustained metabolism in the microcompartments.

**Sustained out-of-equilibrium state in microcompartments**. We first test the experimental conditions under which the NAD-dependent enzymatic reaction is sustained using IMVs (Fig. 3a). The control variables of the system are G6PDH, G6P, NADH and IMVs compositions. We measure the NADH fluorescence over time in 384-well plates under various initial conditions (Fig. 3(b–d)).

First, we vary the G6P concentration and fix all other concentrations. The NADH fluorescence decays to zero in the late kinetics after a plateau of NADH concentration is maintained at intermediate times (Fig. 3b). The decay from the plateau to the background value is universal and all data of the decay collapse when shifted by the plateau time $\tau_1$ (Supplementary Figure 16), proportional to the substrate concentration (Fig. 3b) which means that the reaction is maintained as long as the substrate is present in the system. The timescale of the final decay after the plateau is independent on the initial substrate concentration; This result is expected since at the end of the plateau, the substrate is fully consumed: the decay to zero of the NADH concentration is solely due to the regeneration of NAD+ by the IMVs.

Second, we vary the IMVs concentration and keep the rest constant. In this case, the kinetics shows a plateau for a time function of the IMVs concentration. The kinetics of the final decay is also function of the IMVs concentration (Fig. 3c). Interestingly, both the plateau timescale and the decay timescale are linear in IMVs concentration (Fig. 3c): the kinetics is a universal function when rescaled by a single timescale $\tau_2$, linear in IMVs concentrations (Supplementary Figure 17). The NADH regeneration is therefore limited by the turnover of the IMVs which control the overall kinetics.

Finally, we measure the kinetics by varying the enzyme concentration, fixing all other concentrations (Fig. 3d). In this

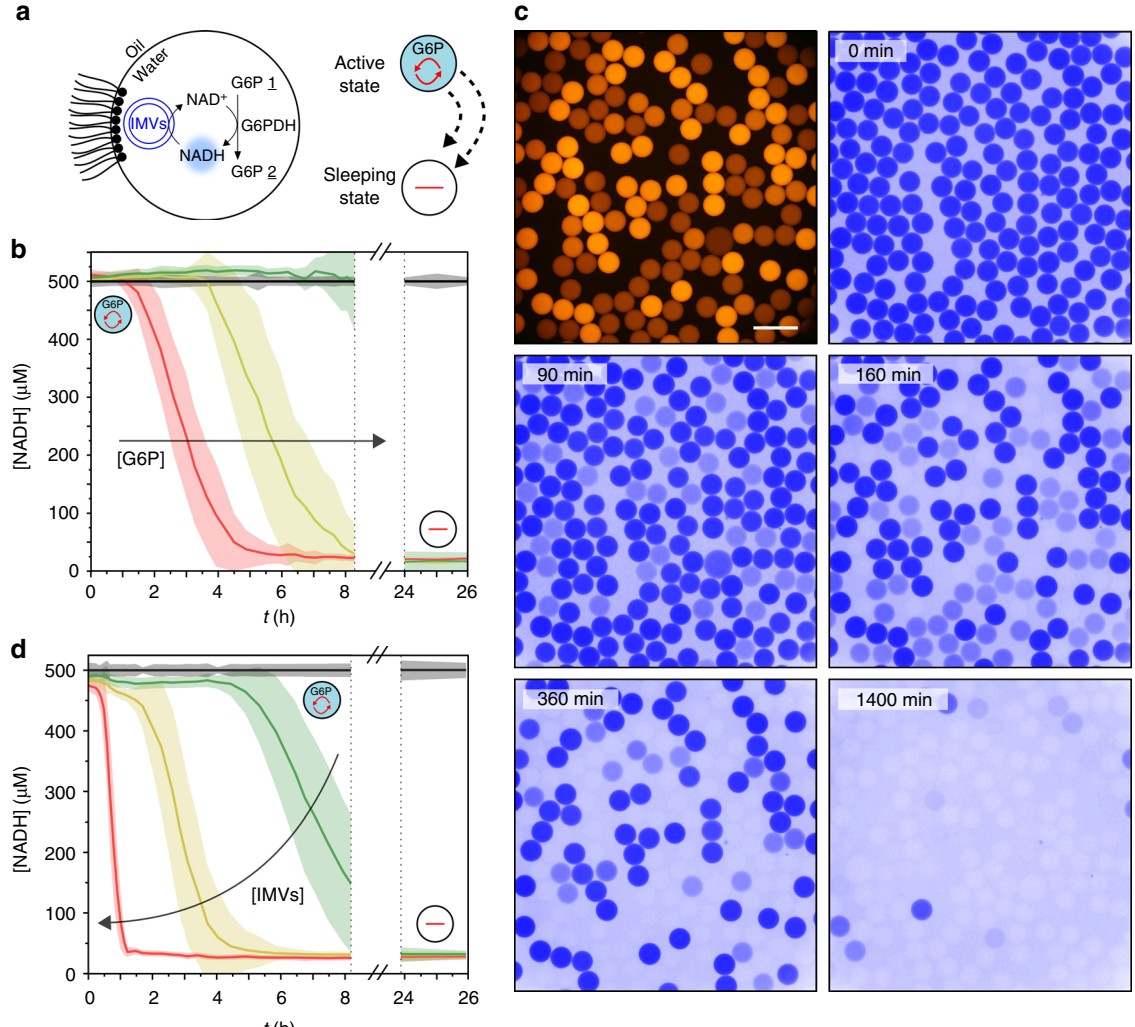

**Fig. 4** Self-sustained compartmentalized metabolism. **a** Graphic view of the compartmentalized network. **b** NADH concentration versus time ($t$) of 30 pL w/o droplets containing NADH (500 μM), G6PDH (0.5 U mL$^{-1}$), IMVs (40 vesicles per droplet) and G6P (0.5 mM (red), 1 mM (yellow) or 2 mM (green)). NADH reference (500 μM) is shown in black. **c** Red and blue fluorescence micrographs of 300 pL droplets containing NADH (1 mM), IMVs (1200 vesicles per droplet), G6PDH (1 U mL$^{-1}$), sulforhodamine B (0, 5, 10, or 20 μM) and G6P (0.5, 1, 1.5, or 2 mM) during 23 h incubation. Scale bar 200 μM. **d** NADH concentration versus time ($t$) of 30 pL w/o droplets containing NADH (500 μM), G6PDH (0.5 U mL$^{-1}$), G6P (1 mM) and IMVs (0 (black), 20 (green), 40 (yellow), or 80 (red) vesicles per droplet). Reactions are performed in NaOH-Tricine buffer (100 mM, pH 8.0) with MgCl$_2$ 5 mM. Error bars are defined as s.d. ($N = 10,000$)

case, the kinetics is bounded by two limiting curves: at high-enzyme concentration, we recover a plateau of sustained reaction and a decay when the substrate is consumed. The enzymatic reaction is fast and as soon as a NADH molecule is regenerated to NAD$^+$ it is reconsumed by the reaction: the kinetics is limited by the IMVs turnover. In the absence of enzyme, the NADH level drops quickly because the only reaction occuring is the IMVs converting the NADH to NAD$^+$. Of course, for intermediate concentration, the balance of reaction rates determines the kinetics: the NADH level reaches a transitory constant level when the two reaction rates balance, with an inflexion point signature.

The system is also sustained from an initial condition where the NADH concentration is initially zero (Supplementary Figure 18). In this case, we observe first an initial increase of the NADH concentration caused by the consumption of the NAD$^+$ by the enzymatic reaction. After this transient state, the system reaches a plateau of NADH concentration independently of the initial NADH concentration: the same out-of-equilibrium state is reached, independent of the initial conditions. This self-sustained

behaviour is also observed with a system involving a two-step reaction. The L-MDH (malate dehydrogenase) activity network (Supplementary Figure 10) is coupled to the NADH oxidation activity of IMVs: the system shows a comparable activity profile, with a plateau of sustained reaction and a decay when the L-malate substrate is consumed (Supplementary Figure 19).

We then integrate the chemical system in our microcompartments (Fig. 4a). Droplets of 30 pL are produced with G6PDH (0.5 U mL$^{-1}$), NADH (500 μM), IMVs (about 40 vesicles per droplet) and G6P at different concentrations (0.5, 1, or 2 mM). We monitor the metabolic state of the microcompartments over 26 h as a function of the G6P initial concentration (Fig. 4b). The microcompartments maintain an out-of-equilibrium active metabolic state thanks to the in situ regeneration of NAD$^+$ until the G6P is fully consumed. The lifetime $\tau_1$ depends on the substrate concentration in the droplets and is comparable to the results performed in bulk experiments (within 1.2-fold difference, Supplementary Figure 20). In addition, we also monitor the metabolic state of microcompartments as a function of G6P

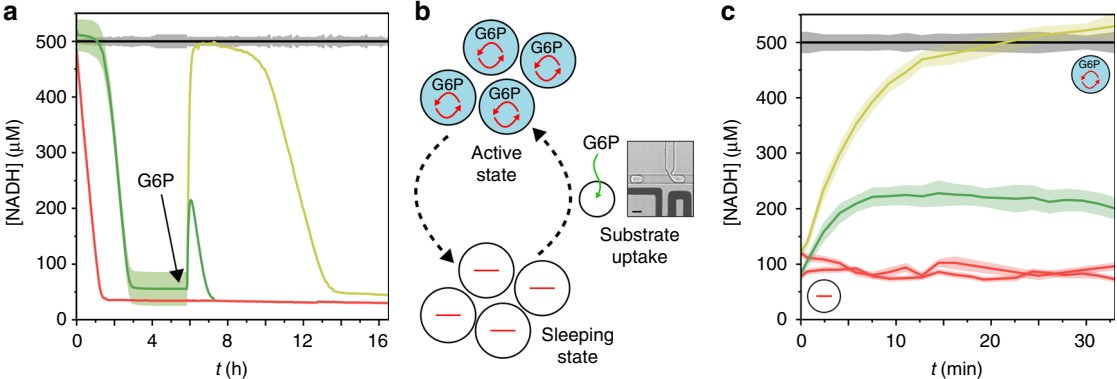

**Fig. 5** Controlling the metabolic state. **a** Bulk kinetics. NADH concentration versus time ($t$) of solutions containing NADH (500 μM), IMVs ($7.2 \times 10^8$ vesicles per mL), G6PDH (0 (red) or 0.1 (green and yellow) U mL$^{-1}$) and G6P (0.5 mM). Control solution without IMVs in black. At $t = 6$ h, addition of 0.2 mM (green) or 2 mM (red, yellow and black) G6P. Error bars are defined as s.d. ($N = 3$). **b–c** Energizing microcompartments. **b** Principle. Compartmentalized self-sustained reactions are maintained out-of-equilibrium in presence of G6P. Microcompartments at the thermodynamic equilibrium ("sleeping state") can be fed with chemical energy by addition of G6P from the surrounding environment (picoinjection). Scale bar 30 μM. **c** NADH concentration versus time ($t$) of 30 pL w/o droplets containing NADH (500 μM), G6PDH (0 (red) or 0.1 U mL$^{-1}$) (yellow and green), G6P (0.5 mM) and IMVs (0 (black) or 20 vesicles per droplet) after 6 h incubation and picoinjection of 0.2 mM (green) or 2 mM (yellow, black, and red) G6P. Error bars are defined as s.d. ($N = 5000$). Reactions are performed in NaOH-Tricine buffer (100 mM, pH 8.0) with MgCl$_2$ 5 mM

substrate concentration within a 4-bit emulsion immobilized in the 2D incubation chamber. We obtain the extinction of the metabolic activity of microcomparmtents at different times in the population, as a function of G6P initial concentration, the more concentrated microcompartments being active the longest (Fig. 4c, Supplementary Movies 1 and 2, Supplementary Figure 21).

The dynamics of the microcompartment population is also controlled by the initial composition of the compartment itself. We prepare a 4-bit population of microcompartments having different amounts of IMVs (0, 20, 40, or 80 vesicles per droplets) with a constant concentration of the substrate. The regeneration of NADH and therefore the rate of the enzymatic conversion depend on the microcompartment in which the reaction occurs (Fig. 4d). The out-of-equilibrium state is maintained over times depending on the IMVs concentration: we have therefore created a population of metabolically active microcompartments having various fitness.

The important message is that—similar to the case of metabolism in living cells—the metabolic system functions under conditions where the cofactor is the limiting compound. With the IMVs regeneration module, we do not depend on a 1:1 stoichiometry of substrate and cofactor; we also do not rely on a perfect balance of the two oxidative states of the cofactors[44]: the stoichiometry is self-controlled in the metabolicaly active state. We produce large populations of independent microcompartments, each having their own reservoir of chemical fuel or their own fitness. As long as the fuel is available the compartment is active. When the fuel is consumed, the microcompartments reach an inactive state. We will refer to this state as a sleeping state. Indeed, while the IMVs are not functional and no reaction occurs in the microcompartments, the regeneration of the cofactor leads to a high concentration of NAD$^+$. As a consequence, the microcompartments should reboot upon the addition of fresh substrate.

**Reactivation of sleeping microcompartments.** Here, we demonstrate that feeding the microcompartments with fresh susbstrate reactivates the metabolic activity. So far our approach has shown that the metabolic state of the microcompartments can be controlled by the experimental conditions. However, living systems make use of the supply of chemical energy by their surrounding environment to maintain their out-of-equilibrium state.

We first demonstrate that the system can be reactivated by an external supply of substrate in bulk experiments. The bulk system is initially in an active metabolic state composed of IMVs, G6PDH, NADH and G6P in 384-well plate (Fig. 5a and Supplementary Figure 22). When all G6P substrate is consumed (0.5 mM), the system reaches a sleeping state characterized by the absence of fluorescence related to a low NADH concentration. We reactivate the system and switch it from the thermodynamic equilibrium sleeping state to the out-of-equilibrium active state upon the addition of G6P (Fig. 4a). For a small amount of G6P added (0.2 mM), the system shows a metabolic pulse—characterized by a brief increase of fluorescence—with the transient consumption of the added substrate. At larger concentrations (2 mM) a sustained active metastable metabolic state is recovered for a few hours. In the latter case, the G6P is metabolized until full consumption and the equilibrium sleeping state is reached again.

To demonstrate that the microcompartments can also be reactivated, we integrate an injection mechanism on-chip to control the addition of fresh substrate molecules in the microcompartments (Fig. 5b). The microcompartments are first produced in an active state and incubated for 6 h until they reach the sleeping state with the complete metabolization of the G6P substrate. The microcompartments are then fed with fresh substrate by targetted picoinjection of either 0.2 mM of 2 mM of G6P substrate. We measure their metabolic activity for about 30 min on-chip (Fig. 5c and Supplementary Figure 22). In both cases, the self-sustained metabolism is reactivated with either a partial metabolic recovery at low concentration (0.2 mM) or a full metabolic activity for high concentration of substrate (2 mM), consistent with the bulk experiments.

## Discussion

We produce and control microcompartments which display a controllable and self-sustained metabolic activity. For the sake of clarity, we want to point out that the activity is self-sustained as long as an energy source, in the form of the substrate, is present in the system and consumed. In this respect, our system is dissipative and maintained out-of-equilibrium for a finite time. First, we design a microfluidic platform to produce large populations of microcompartments in the form of picoliter water-in-oil droplets. Our platform provides statistically relevant data through quantitative and non-invasive fluorescence measurements of the NADH level. We use this high-throughput methodology as a quantitative measurement of the metabolic state of each individual

microcompartment in populations. Our method is generic and usable in a wide range of NAD-dependent enzymes, provided that the chemicals do not optically interfere with the NADH fluorescence.

We chemically functionalize the microcompartments through the bottom-up integration of elementary NAD-dependent metabolic reactions coupled to IMVs for $NAD^+$ regeneration. The reaction is maintained out-of-equilibrium for hours until full consumption of the substrate. Oxygen is an essential element in the reaction (Supplementary Figure 12 and Supplementary Note 3). The solubility of oxygen in fluorinated oils ranges from 14 to 28 mM and is 1.3 mM in water[53]. At chemical equilibrium, the concentration of oxygen in the oil phase is at least ten times larger than the concentration of oxygen in the water droplet. In addition, we produce droplets with a volume ratio of 3 or 1.5(oil):1 (water). The oil therefore acts as a large reservoir of oxygen that constantly replenishes the amount consumed by the reaction in the aqueous phase. We have shown previously that chemical equilibrium in water-in-oil emulsion is reached within seconds even for larger organic molecules, several orders of magnitude faster that the kinetics of the reaction[38]: the kinetics of oxygen transport at the droplet scale is not a limiting factor. As a note, this particular effect is used in other contexts to control polymerization[54] or to improve fermentation processes[55]. The kinetics of the reaction and the concentration profiles inside the compartments are recovered in a minimal kinetic model (Supplementary Figure 23, Supplementary Tables 1 and 2 and Supplementary Note 6). We extract from the numerics the oxygen concentration and the product formation. We recover numerically that the oxygen initially dissolved in the oil contributes to the reaction. In the absence of a sufficiently intense flux of oxygen from the oil the reaction would stop. More detailed models for the IMV turnover could be implemented but the elementary modular description of the system already provides a good representation of our system and the system runs under conditions where the waste product does not affect the metabolic reaction.

The metabolic system functions under conditions where the cofactor is no longer the limiting compound and the balance of the oxidative states of the cofactors is self-controlled, alleviating the need to externally control this parameter. We generate a large population of microcompartments each having a different stock of substrate as chemical fuel. In this case, we show that the activity of each individual microcompartment is sustained for times depending on the amount of susbtrate it initially carries. In addition, at the end of the process, the metabolism of sleeping microcompartments is reactivated by reintroducing fresh substrate.

An important experimental result here is that a minimal metabolic activity can indeed be miniaturized in picoliter-sized microcompartments using a modular approach. We not only control the assembly and functionalization of the microcompartment but also quantitatively show that their behaviour is consistent with the bulk activity of the individual constituents. In this respect, the droplet interface does not act as an inhibitor[56] or as a enhancer of the activity[17,57]. Our experimental systems therefore constitute a step toward the controlled assembly of functional protocells and their quantitative analysis. From a metabolic perspective where complex networks of reactions are usually considered our system might appear far from reality. Nevertheless, the purpose of the metabolism is to maintain a sustained out-of-equilibrium state for the microcompartment. This is precisely what we achieve in a minimal system using a single reaction coupled to a regeneration module. In our opinion, this result is a key for the further assembly of metabolically active functional units. The self-sustained metabolic reaction analysed in thousands of picoliter microcompartments reveals that the variability of the metabolic activity between microcompartments

of equal sizes is small—within tens of percent. In biological terms, we produce microcompartments of equal fitness. But we also design more complex and controlled populations of microcompartments of variable fitness. In our experiments, the substrate of the reaction is partitioned preferentially inside the microcompartment. However, using an enzymatic substrate that is exchanged between droplets[38], these microcompartments would compete for the resources as cells would do. Our systems are therefore of direct relevance as model protocells although they are engineered from droplets which might appear—at first—far from a direct biological relevance.

We now discuss in more details the biological relevance of our approach in the context of synthetic biology and origin of life. The idea that microcompartments are essential to living systems is widely accepted[9–11,17,58]. Yet the minimal form that these compartments should take remains under debate. Fox and Oparin for example suggested that a special class of liquid–liquid phase separation called coacervation could produce microcompartments of relevance in the emergence of life[5,6]. Other types of microcompartments in the form of vesicles or liposomes also have a significant relevance[9–11,17]. In our approach, the key property of the microcompartment is its ability to sequester some molecules and be porous to others to maintain both an out-of-equilibrium state and a certain level of identity. This control of transport and sequestration can be achieved by a membrane or more simply using phase partitioning as in the case of droplets[38]. Membrane-based microcompartments—for example stabilized by lipid bilayers—have the drawback that during growth and division, both the volume and the membrane must grow[10]. For droplets, the volume growth is directly linked to the surface growth and a single process is therefore sufficient for proliferation. Droplets do have a biological relevance: liquid structures are found in living cells in the form of P-granules formed by phase separation in liquid–liquid system providing means to compartmentalize reactions in the cytoplasm[59]. Considering that life has emerged from the most simple system, droplets appear to be relevant models. Using the conceptual basis that the systems found in the laboratory need not be chemically similar to the actual molecular assemblies of living cells[10] but that the key point is to mimic the functions and the essential properties of living systems, our droplets engineered from a phase separation in a fluorinated oil/water mixture are direct analogon of the coacervate droplets and therefore bear a relevance in the context of the build-up of minimal functional microcompartments having life-like properties.

Our experiments provide the basis to chemically functionalize large populations of microcompartments with metabolic activities that can be assembled in a bottom-up approach. A subsequent step that can be envisioned deals with the integration of our modules in the form of other types of microcompartments. Vesicles[35], polymersomes[60], and coacervates[61] would provide interesting alternatives to the droplets for a better control of uptake and release of substrates and products. The control of transport of reagents by phase partitioning in surfactant solutions[38,62] is also an option to deliver substrates and extract waste products. A challenging but interesting route would be the coupling of the chemical functionalization achieved here with a mechanical function to design active micro-systems with life-like properties, such as self propulsion[63] or division[20].

## Methods

**Chemicals.** Fluorescein sodium salt (Sigma, 46960), NADH (Sigma, N8129), NAD$^+$ (Sigma, N6522), ATP (Sigma A26209), glycerol (Sigma, G5516), sulforhodamine B sodium salt (Sigma, S1402), glucose-6-phosphate (Sigma, G7250), FDG (Sigma, F2756), glycerokinase from *Cellulomonas sp.* (Sigma, G6142), glycerol-3-phosphate dehydrogenase from rabbit muscle (Sigma, G6751), glucose-6-phosphate dehydrogenase from *L. mesenteroides* (Sigma, G8529), β-galactosidase from *E. coli* (Sigma, G6008), citrate synthase from porcine heart (Sigma, C3260), L-(-)-malic

acid (Sigma, 02288), Dextran-Cascade Blue (3000 MW) (Molecular Probes, D7132), acetyl-CoA (Roche, 10101907001), L-malate dehydrogenase from pig heart (Roche, 10127914001) solutions were prepared by dissolution in millipore water, NaOH-Tricine buffer (100 mM, pH 8.0, MgCl$_2$ 5 mM), NaOH-glycine buffer(100 mM, pH 9.0) or KOH-Tricine buffer (100 mM, pH 8.0).

**Microfluidic device fabrication.** Devices were made of poly-(dimethylsiloxane) (PDMS, Sylgard 184) from SU8-3000 negative photoresist (MicroChem Corp) molds (20 or 70 µm depth) produced using a soft-lithography procedure as per standard techniques[64]. The surfaces of the microfluidic channels were treated using fluoro-silane (Aquapel, Aquapel) before use.

**Microfluidic device operation.** Either Nemesys syringe pumps (Cetoni) or a pressure driven pump (Fluigent, MFCS-4C) were used to control the flows in the microfluidic devices. Devices were connected to flow controllers with PTFE tubing (Fisher Scientific) with an inner diameter (ID) of 0.3 mm and an outer diameter (OD) of 0.76 mm. Droplets were produced in fluorinated oil (Novec7500, 3 M) and stabilized against coalescence by a perfluoropolyether-polyethyleneglycol block-copolymer surfactant (PFPE-PEG-PFPE), synthesized as previously described[65]. All microfluidic devices were used at controled room temperature (20 °C). Three microfluidic workflows were implemented depending on the experimental needs:

Workflow 1: Short-term multiplexed kinetics (Supplementray Figure 1). A total of 30 pL droplets are produced by the parallelized flow-focusing of two aqueous solutions (200 µL h$^{-1}$) with the fluorinated oil containing 3 wt% of surfactant (600 µL h$^{-1}$). Droplets are produced at 1.9 kHz at each production nozzle and are collected in a glass vial. Several emulsions can be produced sequentially and combined in the same collection vial. Droplets are then reinjected in the kinetics module. Droplets are co-flown (0.5–0.8 bar) with fluorinated oil containing 3 wt% of surfactant (0.5–0.8 bar) and picoinjected with an aqueous phase (0.5–0.8 bar, injected volume: from 4 to 6 times dilutions, 6–10 pL ± 6%) by applying an AC field (20 kHz, 100 V$_{pp}$). A fraction of fluorinated oil was extracted ($-20$–50 µL h$^{-1}$) and the droplets are incubated on-chip in a delay line. Droplet fluorescence is measured at different time points along the delay line up to about 1 h, corresponding to a tenfold increase in the maximal incubation time compared to previously reported systems[41].

Workflow 2: Long-term multiplexed kinetics (Supplementray Figure 2). A total of 30 pL droplets are produced by flow-focusing two aqueous solutions (100 µL h$^{-1}$ each) with the fluorinated oil containing 3 wt% of surfactant (300 µL h$^{-1}$). Droplets are produced at 1.9 kHz and are collected in a glass vial on ice to freeze the metabolism. Several emulsions can be produced sequentially and combined in the same collection vial. Droplets are then warmed up to room temperature and reinjected in a simple reinjection module for fluorescence analysis. Droplets (25 µL h$^{-1}$) are co-flown with fluorinated oil (150 µL h$^{-1}$) to be spaced for droplet fluorescence measurement. Droplets are this way continuously analyzed over hours.

Workflow 3: Short-term single kinetics (Supplementray Figure 3). The microfluidic device is fully integrated. A total of 90 pL droplets are produced by flow-focusing two aqueous solutions (70 µL h$^{-1}$ each) with the fluorinated oil containing 3 wt% of surfactant (200 µL h$^{-1}$). Droplets are produced at 450 Hz. A fraction of fluorinated oil was extracted ($-20$–50 µL h$^{-1}$) and incubated on-chip in a delay line. Droplet fluorescence is measured at different time points along the delay line.

**Fluorescence measurement and data processing.** The optical setup is similar to that reported previously[40] and is detailed in Supplementary Figure 4. Data acquisition (DAQ) and control were performed by a DAQ card (National Instruments) executing a program written in LabView (National Instruments). The data acquisition rate for the system was 200 kHz.

**Microtiterplate fluorescence measurements.** Experiments were performed in 384-well plates (Thermo Fisher) in 45–50 µL of solution. The fluorescence was monitored at room temperature using a spectrofluorometer (SpectraMax Paradigm, Molecular Devices).

**Time-lapse imaging.** Images were taken with a digital camera (Canon, EOS D600). A light emitting diode (365 nm, 1150 mW, Thorlabs) combined to an epifluorescence cube composed of an excitation bandpass filter (F39–370, AHF), a beamspliter (F38–409, AHF) and an emission bandpass filter (F39–438, AHF) were used for the excitation of NADH.

**Extraction and purification of inverted membrane vesicles.** The extraction and purification of Inverted Membrane Vesicles from *E. coli* were done by disintegration of the bacterial membrane and subcellular fractionation. The procedures are fully described in Supplementary Note 3. Briefly, *E. coli* (MG1655) were grown in LB medium and collected by centrifugation. The cells were lysed and homogenized either by ultrasonication (Digital Sonifier Model 450, Branson Ultrasonics Corp.) or French press (Emulsi-Flex C5, Avestin). The membrane was isolated by three ultracentrifugation (Ultrazentrifuge Optima XPN 100, Beckman Coulter) steps (details are given in Supplementary Note 2). Next a density gradient centrifugation in

a sucrose gradient from 20 to 50% was done for 24 h at 240,000×g. The IMVs were located between 35 and 45% sucrose, collected and diluted to 1:4. To concentrate the IMVs, another ultracentrifugation step was performed for 2 h at 433,000×g. Then the pellet containing the IMVs was resuspended in buffer to a concentration of 2 g mL$^{-1}$ and the suspension was filtered with a 0.22 µm sterile filter and frozen at $-80$ °C.

**Characterization of inverted membrane vesicles.** Concentration and size distribution of the vesicles were determined using tunable resistive pulse sensing (TRPS) on a qNano device (Izon Science, Christchurch, New Zealand). For the measurement a NP200 stretchable nanopore was used, which was calibrated with carboxylated polystyrene beads (mean size 350 nm). The lower fluid cell was filled with 80 µL membrane buffer and 30 µL of the vesicles, diluted with membrane buffer, were added to the upper fluid cell. Size distribution and concentration were calculated from the measurement data with the instrument software (Izon Control Suite 2, Christchurch, New Zealand). The average concentration was $2.2 \pm 0.4 \times 10^{11}$ vesicles per mL and the average size was $167 \pm 39$ nm (Supplementary Figure 11).

**Data availability.** The data that support the findings of this study are available from the corresponding author upon request.

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

## Acknowledgements

This project is part of the MaxSynBio consortium and all authors acknowledge funding by the consortium. J.-C.B. also acknowledges the financial support by the ERC (FP7/2007-2013/ERC Grant agreement 306385—SofI), by the 'Région Aquitaine' and by the French Government 'Investments for the Future' Programme, University of Bordeaux Initiative of Excellence (IDEX Bordeaux) (Reference Agence Nationale de la Recherche (ANR)-10-IDEX-03-02). The technical support of Lionel Buisson at the Centre de Recherche Paul Pascal (Pessac, F) is warmly acknowledged. J.-C.B. and T.B. also acknowledge fruitful discussions with Nicolas Bremond and Jean Baudry (ESPCI, Paris; F) for the assembly of 2D incubation chambers and with Yves Gibon (INRA, Villenave d'Ornon; F) for the enzymology of the systems. We are thankful to Tobias Erb (MPI for Terrestrial Microbiology, Marburg; D) for his critical comments and inputs on the manuscript.

## Author contributions

T.B. performed research on microfluidics and integration of IMVs in droplets and contributed to the design of the research under the supervision of J.-C.B.; D.K., C.B., and C.K. performed research on IMVs preparation under the supervision of I.I. and T.V.-K.; C.W. performed the numerical simulations. T.B. and J.-C.B. analyzed data and wrote the paper with contributions of all authors; T.V.-K., K.S., and J.-C.B. designed research.

## Additional information

**Competing interests:** The authors declare no competing interests.

