## [Peer Review File · Nature Communications]

Reviewers' comments:

Reviewer #1 (Remarks to the Author):

This is an interesting work on construction of cell-like compartments based on microfluidics and w/o droplets. The work has two main elements of novelty, namely (1) the use of active enzymes of the cellular respiration in form of E.coli IMVs, and (2) the use of microinjection to "refill" exhausted droplets with fresh substrate.

Certainly the work should have been much more important if vesicles were used instead of w/o droplets, but it is anyway of certain interest for the scientific community, especially for the bottom-up synthetic biology arena.

Interesting aspects

- Microfluidics-generated homogeneous population of compartments
- use, for the first time, of membrane enzymes of the respiratory chain in form of enzymatically active IMVs. These intra-droplet vesicles simulate cellular organelles.
- Quantitative characterization of the "modules" used, as requested by synthetic biology standards, and their combination in modular way in a system of higher complexity

Overall, my evaluation of this work is positive, with some reservation only for the conceptual novelty which is not well evident.

However, the manuscript is well-written, well commented, the figures help understanding the work done, and the scientific message is clear.

There are however some aspects that the Authors should consider in their revision:

- 1) The Authors should specify that w/o droplets, although useful cell-like systems, are not the best ones. In particular the droplet boundary poorly resembles biological cell membranes and thus the relevance of this model, as cellular model, is partial.
- 2) The Authors should better define the conceptual advancement behind this work – which certainly includes technical progresses. After this work, what do we learn about the ongoing efforts of producing artificial cells? And about the "emergence of metabolism"?
- 3) The word "oxygen" can be found only twice in the manuscript (1 time in the figure caption), but this is an essential reactant for the reaction under study. The manuscript should contain much more information about the oxygen dissolved in the emulsion (oil, water), its rate of diffusion, its rate of usage by the respiratory enzymes, and so on. This is a very important addition that must be done in order to clarify the dynamics under investigation.
- 4) The solubility of oxygen in the fluorinated oil should be indicated.
- 5) In the main text (not only in the Materials and Methods section) it should be clearly stated that w/o droplets are covered by a polymer not by lipids, and thus these w/o droplets are only partially cytomimetic. Consequently, I am not sure that the drawing representing the w/o droplets is truly accurate (the boundary molecule appears as a lipid).
- 6) The title of the paper does not really fit with its content. I doubt that the reactions under investigation can be recognized as "minimal metabolism" by the large majority of scientists.
- 7) It is somehow sad to check the reference list and find no citation of groups that actually pioneered this research (Deamer, Szostak, Luisi, Yomo). This gives to the readers a completely distorted reconstruction of historical reality.
- 8) It is not clear what is the "error" in the delivered amount of injected substances. How reproducible is the injection? What is the contribution of it to the overall variance?
- 9) It is not clear whether the strategy of "multiplex format" (page 5) is a genuine novelty of this work or if it has been proposed before. Please address this question in the manuscript. What is "barcoding" why it has been used. Please specify (for example, in the Material/Method section).
- 10) More in general, it is not immediately evident if what is shown here as technical improvements of microfluidics droplet generation / high throughput reading is new or "borrowed" by previous

studies

11) On page 9, line 2: "the experimental error" maybe is not the best term, as it is not a measure of uncertainty in fluorescence reading, rather the variability between the droplets

12) It is important, when commenting data and composition of the systems, to insert the concentrations of the reactants (G6P and NAD⁺, for example), so that the reader can understand if one reactant is in excess with respect to the other, and so on. For example this is relevant in section "Sustained out-of-equilibrium state in microcompartments"

13) An important question is about the system containing G6PDH and IMV. The Authors have employed G6P and NADH as reactants. NAD⁺ is generated by IMV, and oxidises G6P, re-forming NADH. Experiments show that NADH concentration is constant till G6P is present. After its complete conversion to acid, NADH is converted to NAD⁺ at oxygen expenses. What about the alternative design, i.e., G6P and NAD⁺? Did the Authors also try this combination? Why was it discarded?

Reviewer #2 (Remarks to the Author):

In the Manuscript ID: NCOMMS-18-01495 the authors developed high-throughput droplet-based microfluidic platform for the integration and analysis of self-sustained metabolic reactions in water-in-oil microemulsion system. For the analysis, the authors used an assays based on NADH fluorescence to quantify the metabolic state of the microcompartments. The minimal metabolic system was constructed from a reaction converting glucose-6-phosphate (substrate) into 6-phosphogluconactone (product). In this reaction glucose-6-phosphate dehydrogenase and NAD are involved as an enzyme and co-factor, respectively. To sustain the reaction independently of the cofactor stoichiometry, the authors reconstituted within the droplet-based compartments an artificial NAD⁺ regeneration module (i.e., oxidation of NADH back to NAD⁺ in the form of inverted membrane vesicles (IMVs) of *E. coli*). The authors present the robustness of their microfluidic approach for quantitative analysis of microcompartments' metabolic state, with the ability to control on-and-off metabolic activities.

For my opinion, the novelty of this interdisciplinary research lies in the concept that one can generate and precisely control large populations of microcompartments "synthetic cells" with metabolic activities by a bottom-up approach. The biochemistry concepts and microfluidic technology as presented in this research are not novel, but their intelligent integration will influence thinking in the synthetic biology field. The manuscript is well written, the experimental details are in general well described (see minor comments) allowing the researchers in the field to reproduce the work.

These results shall be considered of great interest to the biological community in general and for the synthetic biology applications in particular. I have few minor comments that I would like to bring the attention of the authors:

1. The authors present three well-defined droplets-based microfluidic platforms. However, it is not clearly stated for what purposes the different platforms have been designed. Please clarify the difference in implementation.
2. The authors may present the metabolic activity using glutathione reductase from *Allochromatium vinosum* (reclassified, formerly known as *Chromatium vinosum*) as an alternative NAD⁺ regeneration module.

Reviewer #3 (Remarks to the Author):

In their manuscript 'Minimal metabolism in populations of microcompartments' Beneyton and coworkers use a sophisticated microfluidics system to create large populations of monodisperse emulsion droplets that encapsulate minimal metabolisms with fluorescent readouts. Microfluidics has been used since about twenty years to miniaturize biological studies. Some groups are bringing the microfluidics technology to a new category with an impressive level of sophistication, quality, and control. It is the case in this work. It is superbly executed, quite clean, very well controlled and reproducible. Tens of thousands of emulsion droplets can be analyzed to generate robust statistics of enzymatic reactions carried out in vitro for example. The work is quantitative, which is another strong point. While the hardware platform is powerful, the biology component of the work is less convincing, however. The metabolisms emulated in the droplets are very minimal and the enzymatic reactions are very basic and known for decades. Beyond the impressive technical achievement, it is really hard to find anything new on the biology side in this work. What is the point of developing a superb and powerful microfluidics system to recapitulate enzymatic reactions that have already been used and well described? In addition to this, the work consists mostly of demonstrating that the observations made in large reaction volumes (on well plates) are similar to the observations made in the emulsion droplets.

Some comments that could be used to improve the manuscript.

Major comments:

- As it is, the biology part of the work is weak. It would be good to demonstrate that one can perform new biology or that the experimental system allows achieving what we cannot achieve in large reaction volumes.
- Because some of the metabolisms are relatively simple, adding a model to fit the kinetics would strengthen the work.

Other comments/suggestions:

Title:

- The title is too vague with respect to the work. The title should be closer to the scope of the experiments (metabolism based on NAD/NADH conversion and emulsion droplets as compartments).

Abstract:

- Self-sustained (in abstract and other places in the text): this word seems way too strong because it is simply not self-sustained. Adding a component to regenerate one of the co-factors and extend the metabolism does not mean that the metabolisms are self-sustained. As we see, at some points the signal drops in all the cases.

Text:

- "... with universal readout of the metabolic state of microcompartments using NADH fluorescence.": is it sure that NADH can be used as a universal readout? Because the metabolisms are rather simple in this work and carried out in buffered solutions, NADH fluorescence is practical. As we want to achieve more and more complex enzymatic reactions with potentially many others components, can NADH be used as a readout practically?
- "... of the reaction depends linearly on the"
- Explain how you get the +/- 15% in the fluctuations of the vesicles concentrations.
- "The decay at longer times depends on the substrate".
- Defining the droplet as micron-sized is a bit misleading.
- "... can be extended to a large variety of reactions": a statement that seems again too strong. I suggest the authors be more careful. Because the metabolisms are rather simple, it is not clear how the system presented in this work can be used to study real complex metabolisms that are more relevant to biological systems.
- Most of the discussion section is a summary of the work. It is not really necessary. Instead, it

would be useful to discuss the advantages and limitations of the system, what would be the next step to show that the system brings new capabilities to study biological systems.

Methods:

- Chemical: please give the source of the chemicals listed in the first 3 lines as well as product numbers.
- Microfluidic device fabrication: what is the Sylgard #?
- Microfluidic device operation: at what temperature was the system used?
- "Microtiterplate fluorescence measurements".
- Extraction and purification of inverted membrane vesicles: provide the model and product number for ultrasonication and french press types of equipment and the settings of operation. 54000 rpm: for how long? 2 g pellet / ml: be clearer about what it means.

Figures:

- Some of the figures are hard to see, especially all the schematic of the enzymatic reactions (1a, 1b, 2a) and of the microfluidics (for example SI 1, 2, 3).
- Figure 5: in large reaction volumes the kinetics are shown for 16h, in the emulsion droplets it is shown for 30 minutes. Any reasons for that? Is it possible to show the same sequence of G6P consumption and addition?

Supplementary information:

- The legends of S10 and S11 have been inverted.
- Some of the SI figures could have been more exploited. They are here to support the work, but a minimum of discussion would be useful. For example S11, how do we explain the kinetics and the time to reach plateau? In S14b, I have a hard time reconciling the scales on the 2 pictures, are you sure it's only a factor of 2? S18, in the legend there are 2 (b).
- SI note 4: 2.5 μ l IMVs were added: what is the concentration of IMVs?
- SI note 6: ImageJ 1.x: $x = ?$

Other comments:

- Showing the spectrum (Ex/Em) of the fluorophores used in this work would help.
- One aspect that is not really discussed is the importance of interface effects. A number of groups have reported that the oil/water interface can have a dramatic effect on the enzymatic reactions in emulsion droplets. It would be useful to add a few lines about this, although it does not seem a problem in this work.

Reviewer #1 (Remarks to the Author):

This is an interesting work on construction of cell-like compartments based on microfluidics and w/o droplets. The work has two main elements of novelty, namely (1) the use of active enzymes of the cellular respiration in form of E.coli IMVs, and (2) the use of microinjection to “refill” exhausted droplets with fresh substrate. Certainly the work should have been much more important if vesicles were used instead of w/o droplets, but it is anyway of certain interest for the scientific community, especially for the bottom-up synthetic biology arena.

Interesting aspects:

- Microfluidics-generated homogeneous population of compartments
- use, for the first time, of membrane enzymes of the respiratory chain in form of enzymatically active IMVs. These intra-droplet vesicles simulate cellular organelles.
- Quantitative characterization of the “modules” used, as requested by synthetic biology standards, and their combination in modular way in a system of higher complexity

Overall, my evaluation of this work is positive, with some reservation only for the conceptual novelty which is not well evident. However, the manuscript is well-written, well commented, the figures help understanding the work done, and the scientific message is clear. There are however some aspects that the Authors should consider in their revision:

We appreciate the very positive feedback of the referee and are satisfied to see that our main message is clear. We confirm that the three aspects mentioned by the referee are the interesting points we wanted to convey.

We also appreciate his / her additional comments that we address below and use to now improve our manuscript.

1) The Authors should specify that w/o droplets, although useful cell-like systems, are not the best ones. In particular the droplet boundary poorly resemble biological cell membranes and thus the relevance of this model, as cellular model, is partial.

[Our reply] We appreciate the comment of the author. In our opinion all minimal systems have a partial relevance to model cells as we know them. Even liposomes are partial models as no living organisms is currently compartmentalized within a simple double layer.

The hypothesis that microcompartments based on lipid bilayers are important in the early stage of life evolution is not the sole hypothesis that can be made. In principle, other types of microcompartments can do the trick: the key ingredient for the compartment is its ability to sequester some molecules and be porous to others to maintain both an out-of-equilibrium state and a certain level of identity. This control of transport and sequestration can be achieved by a membrane or more simply using phase partitioning as in the case of droplets. It should be noted that droplets are actually relevant models for the origin of life, for example in coacervates as proposed by Oparin (see also Zwicker *et al. Nature Physics* 2015). These liquid structures are still important in the living systems as we know them with the P-granules of the cytoplasm (Hyman, Weber and Jullicher, *Annual Review of Cell and Developmental Biology*, 30:39-58 (2014)). In this respect our droplets -- which are liquid structures in another liquid arising from a phase separation in an oil / water mixture -- are direct analogon of the coacervate droplets and therefore bear a relevance in the context of the build-up of minimal functional compartments having life-like properties.

[Implementation in the revised version] We agree with the referee that we should clarify the grounds of our model and have added a paragraph in the discussion about this point.

2) The Authors should better define the conceptual advancement behind this work – which certainly includes technical progresses. After this works, what do we learn about the undergoing efforts of producing artificial cells?

[Our reply] We thank the referee for this comment. We agree that a part of the advancement is the technical control of droplets as out-of-equilibrium compartments. Conceptually, the main idea behind our experiments is to demonstrate the functionalisation of microcompartments to put them in an out-of-equilibrium state. We managed to create thousands of these compartments and quantitatively measure their chemical state. We now have a method to determine if a microcompartment is effectively out-of-equilibrium. We also show that this basic metabolism can be reinitiated when fresh substrate is reinjected in the compartment. Our work is one step in the functionalisation of compartments towards artificial cells. One key point in our opinion is that we provide means and methods to measure the properties of the compartments in real time for a quantitative characterization of the compartment. This measurement step is essential in the construction of artificial cells (see also reply to the comments of the referee 3).

We now further demonstrate that we can create large populations of active microcompartment having different fitnesses: we achieve this principle by integrating different amounts of IMVs in the droplets in a controlled way. By doing so, we generate a complex population of compartmentd that sustain their out-of-equilibrium state depending on their composition. In the long run, such a concept is usable to test competition for ressources in primitive systems. These new data are now presented as Figure 4d.

Finally, we want to point out that our system is simple from a metabolic perspective where complex networks of reactions are usually considered. Nevertheless, the purpose of the metabolism is to maintain a sustained out-of equilibrium state for the compartment. This is precisely what we achieve in a minimal system using a single reaction coupled to a regeneration module. In our opinion, this is a key biological result.

[Implementation in the revised version] We have added a paragraph in the discussion about the construction of population of active compartments and included a new Figure 4d.

And about the “emergence of metabolism”?

[Our reply] The term emergence of metabolism appears in our abstract and is misleading as we do not investigate the emergence of metabolim but rather its *de novo* implementation.

[Implementation in the revised version] We have rephrased the abstract.

3) The word “oxygen” can be found only twice in the manuscript (1 time in the figure caption), but this is an essential reactants for the reaction under study. The manuscript should contain much more information about the oxygen dissolved in the emulsion (oil, water), its rate of diffusion, its rate of usage by the respiratory enzymes, and so on. This is a very important addition that must be done in order to clarify the dynamics under investigation.

[Our reply] We agree with the referee that we should improve this part. We have now added a section in the Supplementary Information discussing the reactions. We used these reactions for a numerical model to address one of the comment of referee 3.

In brief, from the stoichiometry of the reaction, we expect to consum 0.5 mol of oxygen for each mol of substrate converted. We expect that the oxygen supply in the system is not limiting based on the following line of argument: The solubility of oxygen in a series of fluorinated oils is 14 to 28 mmol/L depending on the oil used. For water it is 1.3 mmol/L. (see Solute-Solvent Interactions in Perfluorocarbon Solutions of Oxygen. An NMR Study, Hamza, M.A., Serratrice, G., Stébé, M.-J., Delpuech, J.-J., *Journal of the American Chemical Society* 103, 13, Pages 3733-3738 (1981))

From these values, at chemical equilibrium, the concentration of oxygen in the oil is at least ten times the concentration of oxygen in the water droplet. In addition, we produce droplets with a volume ratio of 3 or 1.5(oil):1(water). The oil therefore acts as a large reservoir of oxygen that constantly replenish the amount consumed by the reaction in the aqueous phase. This is the reason why we are not limited in the reaction by the amount of oxygen initially present in the droplet. Actually, the addition of perfluorocarbons as a second immiscible phase, in which oxygen has

higher solubility, is a known strategy to alleviate oxygen limitations in fermenters (Oxygen transfer enhancement in aqueous/perfluorocarbon fermentation systems: I. experimental observations. Junker BH, Hatton TA, Wang DI. *Biotechnology and Bioengineering* 35, 6, Pages 578-85 (1990)) or in chemical synthesis (K. Krutkramelis, B. Xia and J. Oakey, Monodisperse polyethylene glycol diacrylate hydrogel microsphere formation by oxygen-controlled photopolymerization in a microfluidic device, *Lab Chip* 16, 1457 (2016)). In addition, we have shown previously that chemical equilibrium in water in oil emulsion is much faster than the kinetics of the reaction (Gruner *et al. Nature Communications*, 2016) which means that the transport rate at this scale cannot be a limiting factor. A side result of our numerical simulation indicates that depletion of oxygen can indeed occur if the transport of O₂ from a reservoir is not sufficiently fast. Obviously in microfluidics, we are not limited by this transport from the oil.

[Implementation in the revised version] We agree with the referee that this point is important and we have added a section about this point in the text (discussion) and the corresponding data about oxygen consumption and the numerical solution of the model in the Supplementary Information.

4) The solubility of oxygen in the fluorinated oil should be indicated.

[Our reply] We agree with the comment (see reply to point 3).

[Implementation in the revised version] : see reply to point 3.

5) In the main text (not only in the Materials and Methods section) it should be clearly stated that w/o droplets are covered by a polymer not by lipids, and thus these w/o droplets are only partially cytomimetic. Consequently, I am not sure that the drawing representing the w/o droplets are truly accurate (the boundary molecule appears as a lipid).

[Our reply] This comment is a follow-up of the comment 1) of the referee. We agree that the model is partial and detail this point in the discussion. Considering the drawing, we used a common representation of surfactant molecules having a polar head and two fluorophilic tails. Unfortunately, a phospholipid would also be represented the same way. In terms of Physical chemistry, we can not really distinguish both. For clarity, we have added in the legend and in the main text that the droplets are stabilised by bloc copolymers.

[Implementation in the revised version] The main text and the figure caption have been updated.

6) The title of the paper does not really fit with its content. I doubt that the reactions under investigation can be recognized as “minimal metabolism” by the large majority of scientists

[Our reply] As this comment is also mentioned by referee 3, we agree with this comment and propose the new title: Out-of-equilibrium microcompartments for the bottom-up integration of metabolic functions.

[Implementation in the revised version] The title is changed.

7) It is somehow sad to check the reference list and find no citation of groups that actually pioneered this research (Deamer, Szostak, Luisi, Yomo). This gives to the readers a completely distorted reconstruction of historical reality.

[Our reply] We apologize for this apparent distortion. Our aim was not to hide pioneering contributions to the topic.

[Implementation in the revised version] We have rearranged the discussion to account for the relevant references of these groups, especially in the introduction and in the discussion where we present the relevance of our work (reply to comment 1 and 2).

8) It is not clear what is the “error” in the delivered amount of injected substances. How reproducible is the injection? What the contribution of it to the overall variance?

[Our reply] We apologize for this lack of clarity. The initial picoinjection module was first described by *Abate et al.* (Abate, A.R. and Hung, T. and Mary, P. and Agresti, J.J. and Weitz, D.A., High-throughput injection with microfluidics using picoinjectors. *PNAS*, 107, 19163-19166 (2011)). The injection variability was estimated to be 20 %, with injected volumes ranging from 0.5 to 2 pL in droplets of 20 to 30 pL (10 to 60 times dilutions). Using our optimized picoinjection module, we inject volumes to have a 4-6 times dilution of the injected solution (typically 6 pL in 30 pL). In our case, the injection variability is estimated to be 5-7% using image analysis over 100 droplet, which is similar to the variability found in the work of *Sjostrom et al.* using the same dilution regime (Sjostrom, S.L. and Joensson, H.K. and Andersson Svahn, H., *Lab on Chip*, 2013, 13, 1754). This variability is to be compared to the 15% variability estimated based on IMVs distribution within the droplets.

[Implementation in the revised version] For the sake of clarity, we have now added this information (injected volume and injection variability ranges) in the corresponding Methods section.

9) It is not clear whether the strategy of “multiplex format” (page 5) is a genuine novelty of this work or if it has been proposed before. Please address this question in the manuscript. What is “barcoding” why it has been used. Please specify (for example, in the Material/Method section).

[Our reply] Multiplexing is a capability of the technology that was already used in several papers (from our group and others) as described in *Baret et al.*, *Chem Biol.* 17, 528 (2010); *Pekin et al.* *Lab Chip* 11, 2156, (2011); *Sjostrom et al.*, *Lab Chip*, 13, 1754 (2013) and *Lim et al.* *Biomicrofluidics* 9, 034101 (2015). In all cases, a fluorescent marker is used as a bar code to determine the experimental conditions in each droplet from a simple fluorescence readout. Here we used this option to make sure we have for all experiments a positive and a negative control as well as a test of several experimental conditions.

[Implementation in the revised version] To clarify the origin of this multiplexing we have added the corresponding references in the results section.

10) More in general, it is not immediately evident if what is shown here as technical improvements of microfluidics droplet generation / high throughput reading is new or “borrowed” by previous studies

[Our reply] We apologize for this lack of clarity. The droplet generation and high-throughput reading is now well established for passive microreactors. However, we engineered systems usable to measure the properties of active microreactors. On the technical side, we show for the first time a picoinjection with a one hour incubation line on chip which is more than a 10 fold longer than the previous system (1.5 min incubation in *Sjostrom, S.L. and Joensson, H.K. and Andersson Svahn, H., Lab Chip*, 13, 1754 (2013)). Such an improvement is important for experiments beyond our current system.

[Implementation in the revised version] A sentence was added in the results section to refer to existing methods: ‘The design of the platforms and the measurement systems are adapted from previously reported systems’.

In the methods (workflow 1), we have added the sentence: ‘Droplet fluorescence is measured at different time points along the delay line up to about one hour, corresponding to a ten-fold increase in the maximal incubation time compared to previously reported systems~\cite{Sjostrom2013}.’

11) On page 9, line 2: “the experimental error” maybe is not the best term, as it is not a measure of uncertainty in fluorescence reading, rather the variability between the droplets

[Our reply] We agree with this suggestion and modified the manuscript accordingly.

[Implementation in the revised version] “Experimental error” is changed to “droplet to droplet

variability”.

12) It is important, when commenting data and composition of the systems, to insert the concentrations of the reactants (G6P and NAD⁺, for example), so that the reader can understand if one reactant is in excess with respect to the other, and so on. For example this is relevant in section “Sustained out-of-equilibrium state in microcompartments”

[Our reply] We agree with this suggestion and modified the manuscript accordingly.

[Implementation in the revised version] We added the respective species concentration in the initial state of the microcompartments as requested by the referee.

13) An important question is about the system containing G6PDH and IMV. The Authors have employed G6P and NADH as reactants. NAD⁺ is generated by IMV, and oxidises G6P, re-forming NADH. Experiments show that NADH concentration is constant till G6P is present. After its complete conversion to acid, NADH is converted to NAD⁺ at oxygen expenses. What about the alternative design, i.e., G6P and NAD⁺? Did the Authors also try this combination? Why was it discarded?

[Our reply] We thank the referee for this important comment. The system can indeed be sustained starting from the alternative design mentioned, with initially NAD⁺ instead of NADH. As shown in the additional figure below, in a regime where the system is limited by IMVs turnover, the system reaches the same out-of-equilibrium state starting either from G6P / NADH or G6P / NAD⁺. In the last case, NAD⁺ is converted fast by G6PDH to reach a constant NADH level, same level as the G6P/NADH design. The system then decays when G6P is fully consumed. It's worth to notice that the system decays faster in the case of G6P/NAD⁺ initial conditions as the initial NAD⁺ pool is used to convert G6P in the first phase of the system dynamics.

The fact that the plateau is the same for both NADH initial condition is interesting to mention as it confirms that the system reaches an out-of-equilibrium state independent of the initial conditions.

Additional data: Self-sustained G6PDH activity. (a) G6P and NADH as initial substrates. NADH concentration versus time (t) of solutions containing NADH (250 μM), IMVs (1.3.10⁹ vesicles.mL⁻¹), G6P (0.6mM) and G6PDH (0 (red) or 0.6 (green) U.mL⁻¹). Control without IMVs in black. (b) G6P and NAD⁺ as initial substrates. NADH concentration versus time (t) of solutions containing NAD⁺ (250 μM), IMVs (1.3.10⁹ vesicles.mL⁻¹), G6P (0.6mM) and G6PDH (0 (black) or 0.6 (green) U.mL⁻¹). Control without IMVs in red. (c) Combination of (a) and (b). Reaction are performed in NaOH-Tricine buffer (100 mM; pH=8.0) with MgCl₂ 5 mM.

[Implementation in the revised version] We have added the figure in the Supplementary Information and a comment in main text in the sustained out of equilibrium state section.

Reviewer #2 (Remarks to the Author):

In the Manuscript ID: NCOMMS-18-01495 the authors developed high-throughput droplet-based microfluidic platform for the integration and analysis of self-sustained metabolic reactions in water-in-oil microemulsion system. For the analysis, the authors used an assays based on NADH fluorescence to quantify the metabolic state of the microcompartments. The minimal metabolic system was constructed from a reaction converting glucose-6-phosphate (substrate) into 6-phosphogluconactone (product). In this reaction glucose-6-phosphate dehydrogenase and NAD are involved as an enzyme and co-factor, respectively. To sustain the reaction independently of the cofactor stoichiometry, the authors reconstituted within the droplet-based compartments an artificial NAD⁺ regeneration module (i.e., oxidation of NADH back to NAD⁺ in the form of inverted membrane vesicles (IMVs) of *E. coli*). The authors present the robustness of their microfluidic approach for quantitative analysis of microcompartments' metabolic state, with the ability to control on-and-off metabolic activities.

For my opinion, the novelty of this interdisciplinary research lies in the concept that one can generate and precisely control large populations of microcompartments "synthetic cells" with metabolic activities by a bottom-up approach. The biochemistry concepts and microfluidic technology as presented in this research are not novel, but their intelligent integration will influence thinking in the synthetic biology field. The manuscript is well written, the experimental details are in general well described (see minor comments) allowing the researchers in the field to reproduce the work.

These results shall be considered of great interest to the biological community in general and for the synthetic biology applications in particular. I have few minor comments that I would like to bring the attention of the authors:

[Our reply] We appreciate the very positive feedback of the referee and are very happy to see that our analysis is appreciated. We have taken into account the comments of the referee to improve our manuscript.

1. The authors present three well-defined droplets-based microfluidic platforms. However, it is not clearly stated for what purposes the different platforms have been designed. Please clarify the difference in implementation.

[Our reply] We apologize for the lack of clarity related to the three droplet-based microfluidics platforms used in our work. In the initial submission, the three platforms are described in Figure S1, S2, and S3 and their respective purpose and implementation are shown in Supplementary Note 1. We now present this section differently in our revised manuscript.

In brief, the microfluidic workflow 1 is designed to perform short-term multiplexed kinetics (30-60 min incubations). In that case, droplets are produced, with different compositions barcoded with a fluorescent dye. The multiplexed emulsion is then reinjected in the kinetics module for substrate injection, incubation on-chip and fluorescence analysis at different incubation times.

The microfluidic workflow 2 is designed to perform long-term multiplexed kinetics (incubation time higher than 2 h). In that case, droplets are produced with all the component of the system, with different compositions barcoded with a fluorescent dye. The reaction starts when all the constituents are co-encapsulated. The emulsion is in that case incubated off-chip, in a glass vial, and continuously reinjected and analysed in a reinjection device for time-resolved fluorescence measurements.

The microfluidic workflow 3 is designed to performed short-term kinetics (30-60 min). The system is similar to workflow 1 but with only a 1-bit emulsion (instead of a multiplexed emulsion). This workflow flow is composed of a single microfluidic device. In that case, the droplets are produced

with all the components of the system (the reaction starts after droplet production), droplets are incubated on-chip and fluorescence is measured at different incubation times.

[Implementation in the revised version] For the sake of clarity, we moved the content of the Supplementary Note 1 into the Methods section describing the microfluidic device operation. We also clarified the differences between the three workflows in the manuscript.

2. The authors may present the metabolic activity using glutathione reductase from *Allochromatium vinosum* (reclassified, formerly known as *Chromatium vinosum*) as an alternative NAD⁺ regeneration module.

[Our reply] We agree that the regeneration of NAD⁺ can be performed by other means. In principle we can indeed use any compatible NADH-dependant reaction to balance the NADH/NAD⁺ ratio, as for example the glutathione reductase from *Allochromatium vinosium* mentioned by the referee (Chung et al. Journal of Bacteriology, 1975, 203-211). However, the enzyme requires an additional substrate that is not involved in the metabolic pathway which makes the system more complex (two substrates must be used).

In addition, we favour a modular approach and the use of IMVs is therefore highly relevant. Alternatives based on enzymes in solution might not be as modular as the solution we propose here.

[Implementation in the revised version] N/A

Reviewer #3 (Remarks to the Author):

In their manuscript 'Minimal metabolism in populations of microcompartments' Beneyton and coworkers use a sophisticated microfluidics system to create large populations of monodisperse emulsion droplets that encapsulate minimal metabolisms with fluorescent readouts. Microfluidics has been used since about twenty years to miniaturize biological studies. Some groups are bringing the microfluidics technology to a new category with an impressive level of sophistication, quality, and control. It is the case in this work. It is superbly executed, quite clean, very well controlled and reproducible. Tens of thousands of emulsion droplets can be analyzed to generate robust statistics of enzymatic reactions carried out in vitro for example. The work is quantitative, which is another strong point.

[Our reply] We appreciate the very positive feedback of the referee and are very happy to see that the quantitative side of our analysis is appreciated.

While the hardware platform is powerful, the biology component of the work is less convincing, however. The metabolisms emulated in the droplets are very minimal and the enzymatic reactions are very basic and known for decades. Beyond the impressive technical achievement, it is really hard to find anything new on the biology side in this work. What is the point of developing a superb and powerful microfluidics system to recapitulate enzymatic reactions that have already been used and well described? In addition to this, the work consists mostly of demonstrating that the observations made in large reaction volumes (on well plates) are similar to the observations made in the emulsion droplets.

Some comments that could be used to improve the manuscript.

[Our reply] We agree with the comment of the referee that our metabolism is simple. This point was actually driven by the design of our experiments. We want to show the high level of control over a simple metabolic activity to have a solid ground for the further implementation of more complex reactions or for the coupling of this reaction to other functions.

We want to point out one very important point though: the simple fact that the reaction works in a miniaturized compartment is in itself an important result: we can indeed reliably miniaturize and compartmentalize this reaction and all the constituents in a modular approach. Not only this, but our experiments show that the variability between compartments is negligible for our experimental conditions. This result is strong as it shows that we should not expect a significant fitness variability between the compartments when the system is sufficiently controlled. We believe that we have not sufficiently stressed this point in our initial submission and have now added a discussion about this point in the discussion section to strengthen the biological relevance of our results (see also reply to comment 1 and 2 of the referee 1).

[Implementation in the revised version] We have significantly modified the discussion to account for this comment (in line with our reply to the next comment).

Major comments:

- As it is, the biology part of the work is weak. It would be good to demonstrate that one can perform new biology or that the experimental system allows achieving what we cannot achieve in large reaction volumes.

[Our reply] We do not really agree with the comment that the biology part is weak: the biological system is simple – this is true – but the implementation of a modular approach of a minimal metabolically active compartment is new biology – at least from the bottom-up synthetic biology side. We want to point out that the use of IMVs as a regeneration module is a biologically relevant

progress as pointed out by referee 1 (*'use, for the first time, of membrane enzymes of the respiratory chain in form of enzymatically active IMVs. These intra-droplet vesicles simulate cellular organellae'*).

As mentioned in the comment above, the simple fact that the reaction works in a miniaturized compartment is in itself an important result: we can indeed reliably miniaturize and compartmentalize this reaction and all the constituents in a modular approach. The consequence is that we can create populations of compartments with well a defined fitness (here controlled by the amount of IMVs in the droplet). We have now added a new experiment showing the preparation of complex population of various fitness and we show how each population is maintained out-of-equilibrium for times depending on the IMVs concentration. In the long run, more complex but well controlled populations can be assembled with individuals of different fitness. We believe that this further control over populations can be of interest to the community.

In addition, we want to point out that our system is simple from a metabolic perspective where complex networks of reactions are usually considered. Nevertheless, the purpose of the metabolism is to maintain a sustained out-of-equilibrium state for the compartment. This is precisely what we achieve in a minimal system using a single reaction coupled to a regeneration module. In our opinion, this is a key biological result.

[Implementation in the revised version] We have significantly modified the discussion to account for this comment (also based on the previous comment). We have also added a new Figure 4d to demonstrate the preparation of complex populations having different fitnesses and the corresponding text in the results section.

- Because some of the metabolisms are relatively simple, adding a model to fit the kinetics would strengthen the work.

[Our reply] Modelling the kinetics of the reaction was not our primary focus but we agree with the referee that we can do it. We have now modelled the reaction and the results are presented in the Supplementary Information. We recover with a minimal model the shape of our NADH curves and extract now as well the profiles of concentration of the other species involved in the reaction, namely oxygen and the substrate of the enzymatic reaction. It should be noted that we used the most simple description of the IMV kinetics. This model could be improved to better describe the IMVs. However this would be outside of the scope of the manuscript and because our simple model correctly reproduces the behaviour we measure experimentally, we believe that we understand the essential part of the system. An important point to note is that our model currently breaks down when the transport of oxygen from the reservoir to the droplet is slow. In microfluidics, this is not a problem since the oil acts as a reservoir and provides the required amount of oxygen to the droplet (see also reply to referee 1).

[Implementation in the revised version] We have provided a new section in the Supplementary Information on the model and discuss this point in the main text. Christian Wölfer who contributed to this part is now added to the author list.

Other comments/suggestions:

Title:

- The title is too vague with respect to the work. The title should be closer to the scope of the experiments (metabolism based on NAD/NADH conversion and emulsion droplets as compartments).

[Our reply] As this comment is also mentioned by referee 1, we propose the new title: Out-of-equilibrium microcompartments for the bottom-up integration of metabolic functions.

[Implementation in the revised version] The title is changed.

Abstract:

- Self-sustained (in abstract and other places in the text): this word seems way too strong because it is simply not self-sustained. Adding a component to regenerate one of the co-factors and extend the metabolism does not mean that the metabolisms are self-sustained. As we see, at some points the signal drops in all the cases.

[Our reply] We respectfully disagree with the referee: the only state that can be self-sustained indefinitely is the equilibrium state. A self-sustained state will last until the fuel that maintains it out of equilibrium is consumed. In this respect our droplets are no different than any living system: without an energy source they simply decay to equilibrium.

[Implementation in the revised version] For clarity – although we do not agree with the comment – we have added a sentence on this point in the main text (discussion).

Text:

- “..... with universal readout of the metabolic state of microcompartments using NADH fluorescence.”: is it sure that NADH can be used as a universal readout? Because the metabolisms are rather simple in this work and carried out in buffered solutions, NADH fluorescence is practical. As we want to achieve more and more complex enzymatic reactions with potentially many other components, can NADH be used as a readout practically?

[Our reply] Based on the referee comment, we consider that there is a need to clarify our statement. We have added a sentence explaining that the NAD measurement would of course fail if the reaction implemented involves molecules having fluorescence emission in the NAD detection wavelength. However, we believe that a wide range of reactions can be studied using this principle, especially for systems constructed in a bottom-up approach.

[Implementation in the revised version] We have modified the sentence replacing ‘with a universal readout of the metabolic state of microcompartments using NADH fluorescence’ by ‘with a versatile optical readout of the metabolic state of microcompartments using NADH fluorescence’.

We have added the sentence: ‘Our method is usable on the wide range of NAD dependent enzymes and can be generalized to more complex metabolic systems, provides that the chemicals do not optically interfere with the NAD fluorescence.’ in the discussion.

- “..... of the reaction depends linearly on the”

[Implementation in the revised version] this is corrected

- Explain how you get the +/- 15% in the fluctuations of the vesicles concentrations.

[Our reply] This point is explained in the text. We consider a random distribution of vesicles in the droplets which leads to a Gaussian distribution of the number of vesicles per droplets. The width of the distribution is of order $N^{1/2}$ with N the average number of vesicles per droplets

[Implementation in the revised version] N/A

- “The decay at longer times depends on the substrate”.

[Our reply] We indeed have to clarify our sentence and we apologize for the confusion

[Implementation in the revised version] We replaced ‘The decay at longer times is independent on the substrate concentration, which is consistent with the full consumption of the substrate. When all the substrate is consumed, all the experimental conditions are equivalent and therefore show the same kinetic decay’ by ‘The time-scale of the final decay after the plateau is independent on the initial substrate concentration; This result is expected since at the end of the plateau, the substrate is fully consumed: the decay to zero of the NADH concentration is solely due to the regeneration of NAD by the IMVs.’

- Defining the droplet as micron-sized is a bit misleading.

[Implementation in the revised version] This is now corrected as picoliter droplets

- “..... can be extended to a large variety of reactions”: a statement that seems again too strong. I suggest the authors be more careful. Because the metabolisms are rather simple, it is not clear how the system presented in this work can be used to study real complex metabolisms that are more relevant to biological systems.

[Our reply] We agree with the referee that our method would not be usable to study real complex metabolism. But we want to make clear that this is not our aim. Our aim is to construct simple metabolically active microcompartments. In this respect our method is generic as it is based on a measurement of NADH which is used by many enzymes. And we do therefore believe that our statement is not too strong: our method can indeed be extended to a large variety of reactions.

However, we have modified our statements based on the comment above (see first Text comment above), discussing the limitations of our measurement method. We believe that these changes address this comment as well.

[Implementation in the revised version] see our reply to the first Text comment.

- Most of the discussion section is a summary of the work. It is not really necessary. Instead, it would be useful to discuss the advantages and limitations of the system, what would be the next step to show that the system brings new capabilities to study biological systems.

[Our reply] We agree with the suggestion but feel that we still need to keep a summary of our key results.

[Implementation in the revised version] We have now improved the discussion adding several sections (also based on the other comments of the referee and of referee 1). We believe that our changes now address the comment of the referee.

Methods:

- Chemical: please give the source of the chemicals listed in the first 3 lines as well as product numbers.

[Our reply] We agree with the comment. Here are the requested information:

Fluorescein sodium salt (Sigma, 46960), NADH (Sigma, N8129), NAD + (Sigma, N6522), ATP (Sigma A26209), glycerol (Sigma, G5516), sulforhodamine B sodium salt (Sigma, S1402), D-glucose-6-phosphate (Sigma, G7250), FDG (Sigma, F2756), glycerokinase from Cellulomonas sp. (Sigma, G6142), glycerol-3-phosphate dehydrogenase from rabbit muscle (Sigma, G6751), glucose-6-phosphate dehydrogenase from L. mesenteroides (Sigma, G8529), β -galactosidase from E. coli (Sigma, G6008), citrate synthase from porcine heart (Sigma, C3260), L-(-)-malic acid (Sigma, 02288), Dextran-Cascade Blue (3000 MW) (Molecular Probe, D7132), acetyl-CoA (Roche, 10101907001), L-malate dehydrogenase from pig heart (Roche, 10127914001) solutions were prepared by dissolution in millipore water, NaOH-tricine buffer (100 mM, pH 8.0, MgCl 2 5mM), NaOH-glycine buffer(100 mM, pH 9.0) or KOH-tricine buffer (100 mM, pH 8.0).

[Implementation in the revised version] We have modified the materials and method section to include the list of chemicals.

- Microfluidic device fabrication: what is the Sylgard #?

[Our reply] This is Sylgard 184

[Implementation in the revised version] The method section is updated

- Microfluidic device operation: at what temperature was the system used?

[Our reply] We operate at Room temperature (20°C)

[Implementation in the revised version] The method section is updated

- "Microtiterplate fluorescence measurements".

[Implementation in the revised version] The typo is corrected

- Extraction and purification of inverted membrane vesicles: provide the model and product number for ultrasonication and french press types of equipment and the settings of operation. 54000 rpm: for how long? 2 g pellet / ml: be clearer about what it means.

[Our reply] The procedure is described in the Supplementary Note 2.

[Implementation in the revised version] We have added a note to explain that details are given in Supplementary Information.

Figures:

- Some of the figures are hard to see, especially all the schematic of the enzymatic reactions (1a, 1b, 2a) and of the microfluidics (for example SI 1, 2, 3).

[Our reply] We apologize for the lack of readability.

[Implementation in the revised version] The mentioned figures (Fig. 1, Fig. 2, Supplementary Fig.1, Supplementary Fig. 2 and Supplementary Fig. 3) were modified and are now easier to see. We also checked and increased the readability of all the figures, especially Fig. 3, 4 and 5.

- Figure 5: in large reaction volumes the kinetics are shown for 16h, in the emulsion droplets it is shown for 30 minutes. Any reasons for that? Is it possible to show the same sequence of G6P consumption and addition?

[Our reply] The kinetics over large amount of time are more challenging in microfluidics compared to bulk. We could in principle show the same sequence but this would require to use another microfluidic workflow. In addition, we do not think that this result would bring additional insight to the manuscript.

Supplementary information:

- The legends of S10 and S11 have been inverted.

[Implementation in the revised version] The mistake is now corrected

- Some of the SI figures could have been more exploited. They are here to support the work, but a minimum of discussion would be useful. For example S11, how do we explain the kinetics and the time to reach plateau? In S14b, I have a hard time reconciling the scales on the 2 pictures, are you sure it's only a factor of 2? S18, in the legend there are 2 (b).

[Our reply] The ATP production activity of IMVs is demonstrated in Figure S11 using a luciferase assay. The initial increase of luminescence corresponds to an increase of ATP concentration in the system and increasing the NADH concentration leads to increased rate of ATP production. Once all the NADH in the system is consumed, the ATP concentration is constant corresponding to a plateau visible at low luminescence levels (low ATP concentrations). At higher levels of luminescence (higher ATP concentrations) the luciferase is inhibited by its substrate ATP which is then observed as an apparent decrease of the level of luminescence. We now model the kinetics of the reactions in more details and discuss the data in the supporting information.

Concerning the second part of the question, we checked the scales and they are correct.

[Implementation in the revised version] These points are now corrected. We have added discussion about the IMV kinetics throughout the text and in the Supporting Information.

- SI note 4: 2.5µl IMVs were added: what is the concentration of IMVs?

[Our reply] The solution is at $2.2 \cdot 10^{11}$ vesicles per mL (as mentioned in the main text).

[Implementation in the revised version] The Supplementary Note 4 section is updated.

- SI note 6: ImageJ 1.x: x = ?

[Our reply] ImageJ 1.51k

[Implementation in the revised version] The method section is updated

Other comments:

- Showing the spectrum (Ex/Em) of the fluorophores used in this work would help.

[Our reply] We agree that this information could be helpful to some readers. We have measured the spectra in our buffer conditions (see figure below). The respective spectrum of Fluorescein, NADH, Sulforhodamine B and Dextran CascadeBlue were added to the Supplementary Information file. The manuscript was modified accordingly.

Additional data: Fluorescence spectrum. (a) Fluorescence spectrum of the metabolic activity readout fluorescein ($\lambda_{\text{ex}} = 473 \text{ nm}$; $\lambda_{\text{em}} = 520 \text{ nm}$) and respective barcode Dextran CascadeBlue ($\lambda_{\text{ex}} = 375 \text{ nm}$; $\lambda_{\text{em}} = 450 \text{ nm}$). (b) Fluorescence spectrum of the metabolic activity readout NADH ($\lambda_{\text{ex}} = 375 \text{ nm}$; $\lambda_{\text{em}} = 450 \text{ nm}$) and respective barcode Sulforhodamine B ($\lambda_{\text{ex}} = 532 \text{ nm}$; $\lambda_{\text{em}} = 585 \text{ nm}$). All spectrum are recorded at 20°C in NaOH-Tricine buffer (100 mM; pH=8.0) with MgCl_2 5 mM. The grey boxes are indicating the corresponding PMT wavelength windows for the microfluidic optical set up.

[Implementation in the revised version] We have added a new supplementary figure and a reference to this figure in the method section.

- One aspect that is not really discussed is the importance of interface effects. A number of groups have reported that the oil/water interface can have a dramatic effect on the enzymatic reactions in emulsion droplets. It would be useful to add a few lines about this, although it does not seem a problem in this work.

[Our reply] We agree with the referee that this point is not crucial here.

[Implementation in the revised version] We have now added a short discussion on this point (main text, discussion)

REVIEWERS' COMMENTS:

Reviewer #1 (Remarks to the Author):

It seems to me that the Authors have addressed all points raised previously. Therefore I suggest to approve the manuscript in the current form and accept it for publication.

Reviewer #2 (Remarks to the Author):

The authors successfully answered my comments/concerns, therefore I would recommend this manuscript for publication.

Reviewer #3 (Remarks to the Author):

NA